# Effects of a Peripherally Restricted Hybrid Inhibitor of CB1 Receptors and iNOS on Alcohol Drinking Behavior and Alcohol-Induced Endotoxemia

**DOI:** 10.3390/molecules26165089

**Published:** 2021-08-22

**Authors:** Luis Santos-Molina, Alexa Herrerias, Charles N. Zawatsky, Ozge Gunduz-Cinar, Resat Cinar, Malliga R. Iyer, Casey M. Wood, Yuhong Lin, Bin Gao, George Kunos, Grzegorz Godlewski

**Affiliations:** 1Laboratory of Physiologic Studies, National Institute on Alcohol Abuse and Alcoholism, National Institutes of Health, Bethesda, MD 20892, USA; luis.f.santos29@gmail.com (L.S.-M.); alexa.herrerias@nih.gov (A.H.); george.kunos@nih.gov (G.K.); 2Section on Fibrotic Disorders, National Institute on Alcohol Abuse and Alcoholism, National Institutes of Health, Bethesda, MD 20892, USA; nickzawatsky@gmail.com (C.N.Z.); resat.cinar@nih.gov (R.C.); 3Laboratory of Behavioral and Genomic Neuroscience, National Institute on Alcohol Abuse and Alcoholism, National Institutes of Health, Bethesda, MD 20892, USA; ozge.gunduzcinar@nih.gov; 4Section on Medicinal Chemistry, National Institute on Alcohol Abuse and Alcoholism, National Institutes of Health, Bethesda, MD 20892, USA; malliga.iyer@nih.gov (M.R.I.); casey.wood@nih.gov (C.M.W.); 5Laboratory of Liver Diseases, National Institute on Alcohol Abuse and Alcoholism, National Institutes of Health, Bethesda, MD 20892, USA; yulin@mail.nih.gov (Y.L.); bgao@mail.nih.gov (B.G.)

**Keywords:** cannabinoid, MRI-1867, hybrid ligand, CB1 receptor antagonist, iNOS inhibitor, rimonabant, intracerebroventricular administration, alcohol craving, two-bottle paradigm, drinking in the dark

## Abstract

Alcohol consumption is associated with gut dysbiosis, increased intestinal permeability, endotoxemia, and a cascade that leads to persistent systemic inflammation, alcoholic liver disease, and other ailments. Craving for alcohol and its consequences depends, among other things, on the endocannabinoid system. We have analyzed the relative role of central vs. peripheral cannabinoid CB1 receptors (CB1R) using a “two-bottle” as well as a “drinking in the dark” paradigm in mice. The globally acting CB1R antagonist rimonabant and the non-brain penetrant CB1R antagonist JD5037 inhibited voluntary alcohol intake upon systemic but not upon intracerebroventricular administration in doses that elicited anxiogenic-like behavior and blocked CB1R-induced hypothermia and catalepsy. The peripherally restricted hybrid CB1R antagonist/iNOS inhibitor *S*-MRI-1867 was also effective in reducing alcohol consumption after oral gavage, while its *R* enantiomer (CB1R inactive/iNOS inhibitor) was not. The two MRI-1867 enantiomers were equally effective in inhibiting an alcohol-induced increase in portal blood endotoxin concentration that was caused by increased gut permeability. We conclude that (i) activation of peripheral CB1R plays a dominant role in promoting alcohol intake and (ii) the iNOS inhibitory function of MRI-1867 helps in mitigating the alcohol-induced increase in endotoxemia.

## 1. Introduction

Chronic alcohol consumption poses a serious public health problem in the United States and worldwide. An estimated 8.6% Americans remain addicted to alcohol or drugs and there are 15 million new cases of alcohol use disorder (AUD) each year in the US alone, representing an economic burden of nearly a quarter of a trillion dollars [1,2]. The frequency of drinking has been accelerated during the COVID-19 pandemic [3].

Alcohol dependence has traditionally been viewed as a brain disorder caused by neuroadaptations of the reward circuits to alcohol [4]. Despite efforts to develop effective medications, pharmacotherapy to rebalance central neurotransmission has done little to improve drinking outcomes [5,6]. More recent evidence links alcoholism to peripherally born endotoxemia and gut-derived inflammation [7]. One of the key components of the inflammatory cascade is gut microflora lipopolysaccharide (LPS, also termed endotoxin). Alcohol drinking has been shown to induce gut dysbiosis and bacterial overgrowth, impair intestinal permeability, and increase the translocation of bacterial products from the gut into the systemic circulation in rodents [8,9,10], heavy drinkers [11,12], and healthy individuals [13]. Once in circulation, LPS can bind to Toll-like receptors in the liver and innate immune cells to alter the cytokine milieu in favor of inflammatory species, e.g., TNF-α, interleukin (IL)-1β, and IL-6 [14]. These cytokines may be transported to the brain and other remote tissues causing systemic inflammation and tissue damage [15,16]. The reduction in the intestinal bacterial load or alcohol withdrawal has proven effective in attenuating the severity of inflammation and alcohol dependence [17,18]. Ethanol also up-regulates inducible nitric oxide synthase (iNOS), which leads to the dysfunction of intestinal tight junctions [19], gut leakiness, endotoxemia, and liver injury [20,21,22], effects not seen in iNOS-deficient mice [23].

Recent decades have seen growing interest in exploring the endocannabinoid system (ECS) as a target in the treatment of addiction and AUD, driven in part by the synergistic rewarding properties of alcohol and Δ^9^-tetrahydrocannabinol (THC)—the psychoactive component of marijuana that binds to the same cannabinoid receptors as do the endogenous ligands anandamide and 2-arachidonoyl glycerol [24,25,26]. Early preclinical data focused mainly on the therapeutic potential of the prototypical cannabinoid-1 receptor (CB1R) antagonists that cross the blood–brain barrier. Accordingly, blockade of CB1R has been shown to reduce alcohol consumption [27], its rewarding properties [28,29], and to diminish inflammation in the central nervous system (CNS) [30] and in the gut [31]. However, clinical trials to counteract metabolic obesity with SR14716A (rimonabant), the prototype brain penetrant CB1R receptor antagonist/inverse agonist, showed that it produced serious neuropsychiatric adverse events [32], which halted the therapeutic development of this class of compounds.

We have recently reported that the brain non-penetrant CB1R antagonist, JD5037, representing the second generation of CB1R antagonists, significantly suppresses alcohol preference and proposed that endocannabinoids engage CB1R in ghrelin-producing cells of the stomach to promote alcohol drinking in a manner sensitive to blockade by JD5037 [33]. The ability to influence drinking behavior by CB1R outside the CNS has renewed interest in the therapeutic potential of CB1R antagonists. It also raised questions about the role of central versus peripheral CB1R in controlling voluntary ethanol intake, and whether CB1R antagonists could also be beneficial in mitigating other AUD features, e.g., endotoxemia or gut permeability. To address these aspects, we compared here the effects of rimonabant, JD5037, and the two stereoisomers of a newly developed peripherally active hybrid ligand: *S*-MRI-1867 (CB1R antagonist/iNOS inhibitor) and its enantiomer *R*-MRI-1867 (CB1R inactive/iNOS inhibitor) [34,35] in murine models of alcohol drinking. We found that alcohol intake was significantly inhibited by all CB1R antagonists upon systemic, but not upon intracerebroventricular (i.c.v.), administration, and was unaffected by iNOS inhibition, whereas the two MRI-1867 enantiomers were equally effective at inhibiting alcohol-induced increase in blood endotoxin concentration.

## 2. Results

### 2.1. Central Administration of Rimonabant Inhibits CP 55,940-Induced Catalepsy and Hypothermia in Mice

Acute administration of CB1R agonists, such as CP55,940, induces four behavioral phenotypes including hypothermia, hypoalgesia, catalepsy, and hypomotility, with the latter two being exclusively mediated by central CB1R. Thus, these responses are reversible by oral administration of rimonabant and largely insensitive to non-brain penetrant CB1R antagonists [36,37]. To further document the role of central CB1R in these effects, we tested the ability of i.c.v.-administered rimonabant in antagonizing CP55,940-induced catalepsy and hypothermia.

Intraperitoneal (i.p.) injection of CP55,940 (0.1–3 mg/kg) produced a dose-dependent decrease in body temperature (Figure 1a) and cataleptic behavior (Figure 1b) in wild-type mice. These responses were selectively mediated by CB1R, as CB1R-deficient animals remained completely insensitive to treatment apart from the highest CP55,940 dose (3 mg/kg; Figure 1a,b). The i.c.v. infusion of 2 µg of rimonabant effectively blocked the hypothermic (Figure 1c) and cataleptic (Figure 1d) effects of two different doses of CP55,940.

### 2.2. Central Administration of JD5037 Increases Anxiety-Like Behavior in Mice in the Elevated plus Maze Test

The elevated plus maze tests the natural spontaneous exploratory behavior of rodents in novel environments. This trait can be impaired by genetic deletion of CB1R or its pharmacological inhibition by rimonabant.

To test if the central administration of JD5037 triggers an anxiety-like behavior in mice, 1 µg JD5037 or its solvent was delivered directly into the 3rd ventricle. Figure 2 shows the heatmaps of overall activity in drug- and vehicle-treated groups of animals that were tested in the elevated plus maze (Figure 2a) and activities in individual maze compartments (Figure 2b–g). Thus, during the five-minute run in the maze, control mice traveled an average distance of 1189.0 ± 111.6 cm (Figure 2b, *n* = 6) and moved in the arena at the speed of 4.0 ± 0.4 cm/s (Figure 2c, *n* = 6). In contrast, JD5037-treated animals were significantly less active, covering the distance of 740.5 ± 130.6 cm (Figure 2b, *p* < 0.05, *n* = 6) at the velocity of 2.4 ± 0.4 cm/s (Figure 2c, *p* < 0.05, *n* = 6). Drug-treated animals also stayed much longer in the closed arms (Figure 2d) and moved less frequently between the two closed arms of the maze (Figure 2e) than their vehicle-treated counterparts. They refrained from exploring open arms, which is reflected by the failure to enter open arms (Figure 2f) and explore them (Figure 2g), a clear indication of increased anxiety.

### 2.3. Rimonabant and JD5037 Inhibit Voluntary Ethanol Intake Only via the Peripheral Administration Route

To assess the relative role of central versus peripheral CB1R in the control of voluntary ethanol intake, animals receiving CB1R antagonists were tested in a restricted-access drinking-in-the-dark paradigm. Mice exposed to 20% alcohol for a short period at night tend to drink to inebriation, reflected by high blood levels of ethanol [33]. The i.c.v. infusion of rimonabant in a dose that inhibited hypothermia and catalepsy did not alter alcohol drinking (Figure 3a). In contrast, alcohol drinking was markedly reduced when animals received rimonabant by oral gavage. This effect was CB1R-dependent as it did not occur in CB1R KO mice. This is also reflected by the changes in blood alcohol concentration (BAC) in wt and CB1R KO mice (Figure 3b). 

Likewise, i.c.v. administration of JD5037 in a dose that caused anxiety turned out to be ineffective in reducing alcohol drinking (Figure 3c). The drug effectively reduced alcohol drinking only when given by oral gavage (Figure 3d). This observation is consistent with our earlier finding, which also showed that the effect of JD5037 was CB1R-dependent as it did not occur in CB1R-deficient mice [33].

### 2.4. MRI-1867 Regulates Alcohol Consumption in Mice through the Inhibition of Peripheral CB1R but Not iNOS

The effect of MRI-1867 on alcohol intake in mice was isomer-specific in two experimental models. Thus, in the drinking-in-the-dark paradigm, oral administration of the S-MRI-1867 (CB1R antagonist/iNOS inhibitor) inhibited alcohol consumption in a dose-dependent fashion, whereas alcohol intake was unaffected by similar treatment with R-MRI-1867 (CB1R inactive/iNOS inhibitor). The trend is also reflected by the changes in serum acetaldehyde and alcohol levels (Figure 4a), indicating that the drug does not affect the rate of alcohol metabolism. 

The same pattern is evident when using the 2-bottle choice test. Consistent with our earlier study [33], male C57BL/6J mice with continuous access to water and 15% ethanol solution displayed high preference for alcohol (64.2 ± 1.2%), resulting in an average daily intake of 10.3 ± 0.2 mg ethanol/g body weight (*n* = 74). The high alcohol preference and intake remained unaffected by daily gavage with vehicle or R-MRI-1867, whereas S-MRI-1867 was effective in reducing alcohol preference and intake, without affecting total liquid consumption or food intake (Figure 4b).

### 2.5. Inhibition of iNOS by MRI-1867 Enantiomers Decreases Serum Endotoxin Level in Acutely Alcohol-Intoxicated Mice

Acute challenge of mice with ethanol is known to increase gut permeability [9]. We used changes in the amount of endotoxin measured in the portal vein that delivers blood to the liver as an indicator of gut permeability. As expected, endotoxin level was significantly higher in animals exposed to alcohol. Pretreatment of mice with either MRI-1867 enantiomer significantly reduced endotoxin level in the blood (Figure 5).

Two hours after oral administration of 10 mg/kg *S*-MRI-1867, drug concentration measured in different segments of the gastrointestinal tract was on average ~36 μmol/g wet tissue weight (Table 1, *n* = 3).

## 3. Discussion

Evidence has accumulated over the years to implicate endocannabinoids acting via CB1R in the control of alcohol seeking behavior. Increasing endocannabinoid ‘tone’ by genetic deletion or pharmacologic inhibition of enzymes involved in endocannabinoid degradation was reported to increase alcohol intake and preference in rodent drinking paradigms [38]. Conversely, alcohol preference and intake are suppressed by genetic knockout or pharmacologic blockade of CB1R. Specifically, the globally acting CB1R antagonist/inverse agonist rimonabant reduced alcohol preference and intake not only upon systemic administration [27], but also when it was microinjected into limbic structures believed to regulate alcohol drinking behavior, such as the ventral tegmental area (VTA), nucleus accumbens, or prefrontal cortex [39,40,41], providing strong support for the role of these central structures in the control of addictive drinking. Therefore, the recent report that a peripherally restricted CB1R antagonist was as effective as rimonabant in inhibiting alcohol preference and intake in mice was unexpected, as it suggested that alcohol drinking behavior can be disrupted by blocking CB1R at a peripheral site(s) [33]. Further evidence indicated the existence of a gut–brain axis, whereby endocannabinoids acting via CB1R on ghrelin-producing cells in the stomach promote the posttranslational activation of ghrelin and its signaling to the brain via ghrelin receptors on vagal afferent terminals in the stomach [33]. However, the relative contribution of peripheral vs. central CB1R in the control of alcohol-seeking behavior has remained unclear.

In the current study, we have explored the relative role of central vs. peripheral CB1R in promoting alcohol drinking in mice using the two-bottle free choice as well as the drinking-in-the-dark paradigm in mice. We found that the brain penetrant CB1R antagonist rimonabant and its non-brain penetrant counterpart JD5037 significantly inhibited voluntary alcohol intake upon systemic administration. The response was CB1R-dependent as it was absent in CB1R KO mice. It was also reflected by the reduction in inebriating blood alcohol levels in wild-type, but not in CB1R KO mice. However, both drugs lost their efficacy to modulate alcohol drinking when administered intracerebroventricularly, even though they elicited an anxiogenic-like response and blocked CB1-induced hypothermia and catalepsy, indicating that they were able to engage CB1R in the CNS. Thus, our observations do not support a significant role of the central CB1R in the control of alcohol drinking in mice. This conclusion is also compatible with the earlier finding that systemically administered rimonabant lost its ability to inhibit alcohol preference and intake in mice with vagal afferent denervation [33]. However, the input of central CB1R in mediating other symptoms of AUD, e.g., alcohol tolerance, cannot be entirely excluded as it may rest on the experiment model. It has been shown that mice chronically exposed to alcohol display considerably lower sensitivity to cannabinoid-induced hypomotility, hypothermia, and antinociception because of lower CB1R density in the hypothalamus, VTA, and other brain areas [42].

A second objective of this study was to further test the role of peripheral CB1R in the control of alcohol drinking behavior and explore potential additional mechanisms. As detailed in the introduction, chronic alcohol consumption has been associated with low-grade inflammation, including intestinal inflammation, and the resulting increase in gut permeability has been causally linked to increased expression and activity of iNOS in the intestinal mucosa [7,18,23]. Endocannabinoids have also been shown to increase gut permeability via CB1R activation, an effect reversible by CB1R antagonists [31]. The alcohol-induced dysfunction of the intestinal barrier results in the translocation of bacterial endotoxin (LPS) and gut-derived microbial products into the circulation where their presence correlated with increased expression of inflammatory cytokines in peripheral blood mononuclear cells [43]. These changes have been implicated in alcoholic steatohepatitis and were also found to correlate with alcohol craving and consumption by alcohol-dependent individuals [44]. The recent development of a peripherally restricted hybrid inhibitor of CB1R and iNOS, *S*-MRI-1867 has enabled us to assess the relative contribution of these two molecules to alcohol drinking behavior and alcohol-induced intestinal barrier dysfunction in mouse models. The incorporation of acetamidine, a known inhibitor of iNOS, into the side chain of the same ibipinabant scaffold used to generate JD5037, imparted iNOS inhibitory activity to both the *S*- and *R*-enantiomers of MRI-1867, whereas the nanomolar CB1R inhibitory potency uniquely resides in the *S*-enantiomer [34,45]. Similar to rimonabant and the single-target peripheral CB1R antagonist JD5037, orally administered *S*-MRI-1867 potently inhibited alcohol drinking in both drinking models used whereas *R*-MRI-1867 was without such effect, indicating the exclusive role of peripheral CB1R in this effect. In contrast, the alcohol-induced increase in plasma levels of LPS was similarly inhibited by *S*- and *R*-MRI-1867, which strongly suggests the dominant role of iNOS rather than CB1R inhibition in this effect. This is further supported by the fact that the drug concentration measured across the gastrointestinal tract was well above the IC50 value for MRI-1867 enantiomers (≥10 µM) that inhibits iNOS activity in in vitro assays and in mouse tissue homogenates [35,45]. Since both drug enantiomers were used at their maximally effective doses and both were active iNOS inhibitors, we could not assess the relative contribution of iNOS and CB1R components of MRI-1867 to endotoxemia and intestinal permeability, which would require the drug to be administered at submaximal doses. The role of CB1R should be considered in the light of the fact that endocannabinoids increase intestinal permeability in Caco-2 cells [46,47] as well as in obese mice [48], where LPS acts as a master switch to control adipose tissue metabolism, sensitive to blockade by rimonabant. Therefore, a possible crosstalk between iNOS and CB1R in the control intestinal barrier integrity remains to be explored.

In conclusion, our observations support the predominant role of peripheral CB1R in the control of alcohol drinking behavior. Furthermore, our findings using a novel, peripherally restricted, hybrid inhibitor of CB1R and iNOS indicate that engaging these two distinct targets, with respective roles in the drive to drink and alcohol-induced organ toxicity, by a single chemical entity could represent an attractive therapeutic approach to simultaneously mitigate the urge to consume alcohol and some of its harmful peripheral effects. So far, simultaneous inhibition of CB1R and iNOS has been a promising therapeutic strategy for the treatment of pulmonary fibrosis [45], Hermansky-Pudlak syndrome, pulmonary fibrosis [49] liver fibrosis [34], obesity-related dyslipidemia [35], and chronic kidney disease [50], and the hybrid inhibitor featured in all these studies is in early clinical development. The current study emphasizes its potential therapeutic use in AUD and alcohol-induced organ injury [51].

## 4. Materials and Methods

### 4.1. Animals

All animal procedures were approved by the Institutional Animal Care and Use Committee of NIAAA, NIH (Animal Experimentation permit number LPS-GK-1), and the experiments were carried out in accordance with its guidelines. C57BL/6J mice were purchased from The Jackson Laboratory (USA). Cnr1^−/−^ were generated as described [52] and were propagated by heterozygote breeding, using corresponding wild-type littermates as controls. The strain had been backcrossed at least 10 times to maintain the C57BL/6J background. Animals were housed 4 per cage on a 12-/12 h light/dark cycle, had free access to food (rodent sterilizable diet; Harlan Teklad, USA) and water, and were experimentally naïve before testing. Mice were housed individually for the two-bottle alcohol preference and drinking-in-the-dark tests. They were allowed at least 5–7 days to habituate to the experimental conditions and handling prior to testing. 

### 4.2. Cannulation and Intracerebroventricular Microinfusion of CB1R Antagonists

For experiments requiring intracerebroventricular (i.c.v.) drug infusion, pre-canulated (third ventricle) C57BL/6J mice were received from The Jackson Laboratory. Animals had internal guide cannula (2.5 mm long; Plastics One, USA) mounted in the 3rd ventricle (standard coordinates—ML: +1.0, RC: −0.4, DV: 2.0 mm) and protected by a dummy cap until the experiment. For more information on the cannulation procedure, animal care, and use, see the link: https://www.jax.org/-/media/jaxweb/files/jax-mice-and-services/brain-cannulation-information-care-use.pdf?la=en&hash=FD75F73AB0CD7A47808C78D0FC405AB3AF123F3B (accessed on 14 June 2021).

Conscious freely moving mice were infused i.c.v. with rimonabant (2 µg), JD5037 (1 µg), or their solvents. Drugs were applied in a volume of 1 μL over the course of 2 min via 33 g internal injector (P1 Technologies, Roanoke, VA, USA) connected with the 2 µL precision glass Hamilton syringe (USA) by a PE-20 tubing (Fisher Scientific, Hampton, NH, USA). The infusion rate and volume were controlled through the use of the syringe pump (model PHD 22/2000, Harvard Apparatus, Cambridge, MA, USA). Following 2 min infusion, injectors were left in place for additional 3 min to allow a passive diffusion of drugs into the tissue.

### 4.3. Catalepsy and Hypothermia

Catalepsy was assessed using the bar test described [36] with modifications. Mice were removed from home cages and their forepaws were placed on a horizontal stainless-steel rod, 0.5 cm in diameter, positioned 3.5 cm above the bench surface. Cataleptic behavior was defined as the time the animals remained motionless holding on to the bar. Vehicle-treated mice routinely went off the bar within ~2 s. The arbitrary cutoff time for cataleptic mice was 60 s. Hypothermia was then evaluated by measuring core body temperature with a rectal probe (Ellab Inc., Denver, CO, USA).

To determine a working range of CP 55,940 doses that induce catalepsy and hypothermia through CB1R, a dose–response curve for CP 55,940 was constructed first. Pharmacologically naïve Cnr1^−/−^ and wild-type littermates were injected with increasing doses of CP 55,940 (0.1, 0.3, 1, and 3 mg/kg; i.p.) at 30 min intervals. Hypothermia and catalepsy tests were performed every 30 min before ordering the next dose of CP 55,940. Subsequently, conscious freely moving C57BL/6J mice were infused i.c.v. with rimonabant (2 µg), *S*-MRI1867 (3 µg), or vehicle followed by an injection of a single dose of CP 55,940 (0.1 or 0.3 mg/kg, i.p.) 30 min later. The cataleptic behavior and body temperature were evaluated before and 30 min after CP 55,940 injection.

### 4.4. Elevated plus Maze Test

Anxiety-related behavior was assessed using the elevated plus maze (EPM) test as described [53], with modifications. C57BL/6J mice were allowed to acclimate in their home cages for 2 h prior to the procedure. Animals received i.c.v infusion of JD5037 1 µg or vehicle and were tested in the EPM one hour later.

Mice were given 5 min to explore an elevated platform (72 cm above the floor) consisting of two opposing open and closed arms (each 30 × 5 cm) crossing each other. Illumination in the open and closed arms was 20 and 90 Lux, respectively. Mice were placed individually in the center of the platform (5 × 5 cm) facing an open arm. The behavior of each mouse was monitored by a computer-assisted video tracking system EthoVision XT (Noldus, Leesburg, VA, USA). The tested categories include total distance (cm) and average velocity (cm/s) of the run in the entire maze, cumulative duration (% of total time) spent in closed and open arms, number of entries into each closed and open arm of the maze (in-zone frequency). Arm entries were defined as crossing of the center point (located at approximately two thirds of the mouse body) into the arm. Number of entries, time spent in the open or closed arms, and distance travelled were measured by an automated HindSight software system (Hindsight, version 1.4, Hindsight Software Solutions Inc., Frisco, TX, USA). 

### 4.5. Two-Bottle Alcohol Preference Test

The procedure was performed as described [27], with modifications. Animals were individually housed and acclimated to the paradigm for 5–7 days by having access to two identical water bottles and handled daily to minimize the stress associated with drug testing. Animals were first subjected to a gradual increase in ethanol concentration in a drinking bottle (3%, 6%, 9%, 12%, 15%), while the other bottle contained water. The position of the bottles was changed every day, and alcohol and water bottles were replaced every 4 days. Once alcohol concentration reached 15%, animals remained on the paradigm for 10 days. Starting on day 2, mice received vehicle by oral gavage for 4 days, followed by a daily treatment with the global or peripheral CB1 antagonist (*S*-MRI-1867 1, 3, 10 mg/kg; *R*-MRI-1867 10 mg/kg,) or vehicle (control group) for another 5 days, one hour before the dark period. All animals were sacrificed for blood and tissue collection 12–16 h after treatment.

### 4.6. Drinking in the Dark

The procedure was performed as described [53], with modifications. Mice were individually housed and acclimated to the room for 5–7 days before testing and randomly assigned to treatment groups. Starting 3 h into the dark cycle, water bottles were replaced with 20% ethanol in 25 × 100 mm glass tubes fitted with metal sippers, Access to the ethanol solution was limited to 4 h every day. One hour before the dark period on day 4, animals received a single dose of CB1R antagonist by oral gavage (*S*-MRI-1867 1, 3, 10 mg/kg; *R*-MRI-1867 10 mg/kg, rimonabant 10 mg/kg; JD5037 3 mg/kg or vehicle) or i.c.v. (*S*-MRI-1867 2 µg; rimonabant 2 µg; JD5037 1 µg or vehicle), and the alcohol session was repeated. Immediately after the drinking session, mice were deeply anesthetized with isoflurane. Blood was collected by traumatic avulsion of the orbital globe and kept for 10 min at room temperature in Eppendorf tubes followed by centrifugation at 3000× *g* for 10 min at 4 °C. Serum was collected and kept frozen in sealed vials at −80 °C until analysis.

### 4.7. Acute Ethanol Intoxication

The acute intoxication was performed on 8-week-old C57BL/6J mice as developed by [54], with modifications. This model was designed to achieve blood alcohol levels that would produce physiological effects comparable to human binge drinking. The animals were fasted overnight. On the day of the experiment, they were transferred to the procedure room and allowed to acclimate to the new environment for 2 h while remaining in their home cages. Animals were divided into six treatment groups. Each mouse received *S*-MRI-1867 10 mg/kg or *R*-MRI-1867 10 mg/kg or vehicle by oral gavage, followed 6 g/kg ethanol (30% *w*/*v*) or saline by the same administration route 30 min later. Mice were anaesthetized by isoflurane 90 min later and sacrificed for blood withdrawal and tissue collection. To determine endotoxin levels, blood was drawn aseptically from the portal vein without anticoagulant and clotted for 30 min at room temperature. Samples were then centrifuged at 2000× *g* for 15 min at room temperature. Serum was pipetted off aseptically into new sterile vials and kept frozen at −80 °C until analysis. For the collection of tissue specimens, the entire alimentary tract was removed and cleaned from the mesentery from mice treated with S-MRI-1867 and saline. Stomach and ~5 cm segments of duodenum, jejunum, ileum, and colon were dissected out, flushed three times with 20 mL ice-cold PBS, and kept in −80 °C freezer till analysis.

### 4.8. Blood Alcohol and Acetaldehyde Assays

Alcohol concentration was measured in serum using a sample analyzer (Model GM7 Micro-Stat, Analox Instruments Ltd., Amblecote, United Kingdom) or an alcohol dehydrogenase kit (procedure 332-UV; Sigma-Aldrich, St. Louis, MO USA) according to the manufacturer’s instructions.

The acetaldehyde measurement by gas chromatography-mass spectrometry (GC/MS) was conducted as described by Jin et al., [55] with modifications. In brief, 50 μL of serum or about 50 mg of liver tissue were mixed with 5 μM of 2H6–EtOH (internal standard for ethanol) and 0.04 μM of ^2^H_4_-acetaldehyde (internal standard for acetaldehyde) prior to adding 200 μL of 0.6 N perchloric acid into each sample. Serum samples were centrifuged at 1780× *g* × 15 min at 2 °C after vortexed for 30 s. Liver samples were homogenized and then centrifuged at 13,200× *g* × 15 min at 2 °C. The supernatant of each sample was quantitatively transferred into a 20 mL headspace vial and capped immediately. Headspace vials were then loaded onto the 111-vial tray of a Headspace Sampler coupled to GC/MS (Agilent Technologies, Santa Clara, CA, USA). The concentrations of acetaldehyde in serum were calculated by comparing the integrated areas of ethanol and acetaldehyde peaks on the gas chromatograms with those of known concentrations of internal standards added in each sample.

### 4.9. Serum Endotoxin Measurement

Serum samples were thawed and diluted 1:5 with sterile water. Samples were then heat shocked at 75 °C in Eppendorf ThermoMixer C (Eppendorf, Enfield, CT, USA) and allowed to cool to room temperature for 10 min prior to colorimetric assay, according to the manufacturer’s instructions (Pierce™ Chromogenic Endotoxin Quant Kit; Thermo Fisher Scientific, Waltham, MA, USA).

All materials used in the assay (e.g., pipette tips, glass tubes, microcentrifuge tubes, and disposable 96-well microplates) were endotoxin-free.

### 4.10. Tissue Levels of MRI-1867

Tissues were extracted as described previously [45]. MRI-1867 concentration was determined by stable isotope dilution liquid chromatography/tandem mass spectrometry (LC-MS/MS) using an Agilent 6410 triple quadrupole mass spectrometer (Agilent Technologies, Santa Clara, CA, USA) coupled to an Agilent 1200 LC system (Agilent Technologies, USA). Chromatographic and mass spectrometer conditions were set as described previously [45]. The molecular ion and fragments for MRI-1867 were as follows: *m*/*z* 548.1→145 and 548.1→257.1 (CID energy: 56 V and 24 V, respectively). The amounts of MRI-1867 in the samples were determined against standard curves. Values are expressed as µmol/gwet tissue weight.

### 4.11. Drugs

The synthesis, purification, and verification of the MRI-1867 and JD5037 structures were performed as described [34]. Rimonabant was obtained through the National Institute on Drug Abuse Drug Supply Program (ref number NOCD-082). CP 55,940 was from Tocris Bioscience (Minneapolis, MN USA). Cell culture-grade DMSO (ATCC, Manassas, VA, USA) was used as a solvent mix for intracerebroventricular microinfusions. Ethanol (ethyl alcohol, U.S.P. 200 proof, anhydrous; The Warner Graham Company, Cockeysville, MD, USA) was obtained through NIH Supply Center. A standard rodent chow (Teklad laboratory animal diet) was purchased from Envigo (Indianapolis, IN, USA). Pierce™ Chromogenic Endotoxin Quant Kit was from Thermo Fisher Scientific (Waltham, MA, USA). All other chemicals were from MilliporeSigma (Rockville, MD, USA). Drugs were aliquoted and stored at −80 °C. For intragastric administration, chemicals were dissolved in DMSO:Tween 80:water (5:2:93). For intracerebroventricular (i.c.v.) microinfusion, drugs were suspended in DMSO:Tween 80:PEG400:saline (solvent composition for JD5037 3:8:30:59 and rimonabant 1:5:30:64).

### 4.12. Quantification and Statistical Analysis

Values are presented as mean ± s.e.m., with the number of replicates and the level of significance reported in figures and figure legends. Statistical data analysis was performed using GraphPad Prism 8 for Windows (version 8.0.1, GraphPad Software, San Diego, CA, USA). A two-tailed Student’s *t*-test for paired or unpaired data was used for comparison of values between two groups. For multiple groups, ordinary one-way ANOVA followed by Dunnett’s post hoc test was applied. Time-dependent variables were analyzed by two-way ANOVA followed by Tukey’s multiple comparisons test. Differences were considered significant when *p* < 0.05.

## Figures and Tables

**Figure 1 molecules-26-05089-f001:**
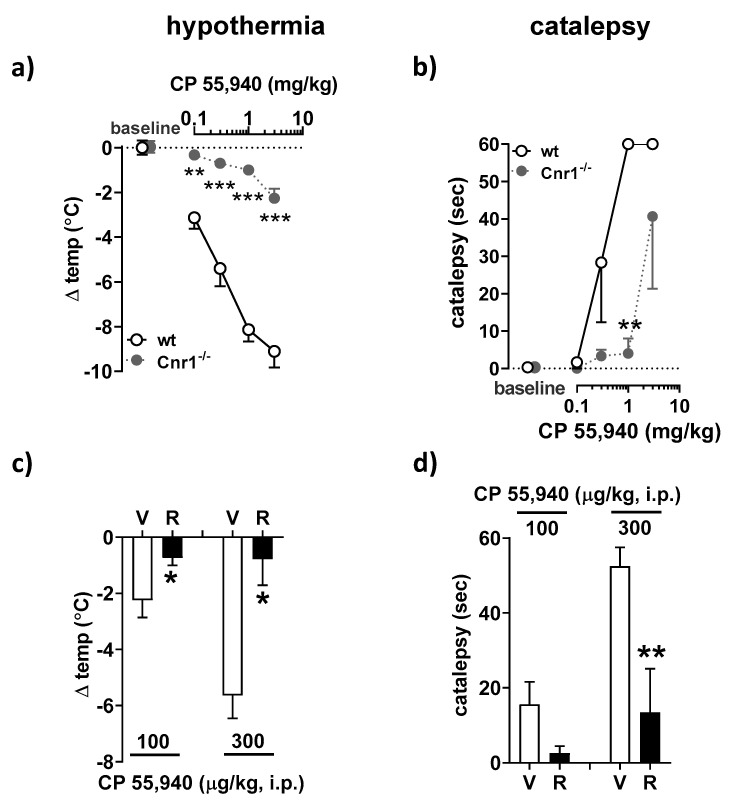
Effect of central administration of rimonabant on CP 55,940-induced catalepsy and hypothermia in mice. (**a**,**b**) Pharmacologically naïve Cnr1^−/−^ (ko) mice and wild-type littermates (wt) were injected with CP 55,940 (0.1, 0.3, 1 and 3 mg/kg; i.p.) at 30 min intervals. Body temperature (**a**) and cataleptic behavior (**b**) were evaluated before administering the next dose of CP 55,940. The respective baseline body temperature in wt and ko groups prior to CP 55,940 injection were 37.1 ± 0.3 °C (*n* = 3) and 37.2 ± 0.3 °C (*n* = 3). Mice held the bar for 0.3 ± 0.3 s and 0.3 ± 0.3 s in wt and ko groups, respectively. (**c**,**d**) Conscious freely moving C57BL/6J mice (*n* = 5 animals per treatment group) were infused intracerebroventricularly (i.c.v.) with rimonabant (2 µg, R) or its solvent (V), followed 30 min later by intraperitoneal (i.p.) injection of CP 55,940 (0.1 or 0.3 mg/kg). Another 30 min passed before the hypothermic (**c**) and cataleptic (**d**) responses were measured. The respective baseline body temperatures before CP 55,940 injection to V- and R-infused groups were 37.6 ± 0.1 °C and 37.5 ± 0.1 °C (*n* = 10). Mice held the bar for 0.3 ± 0.3 s and 0.1 ± 0.1 s in V- and R-treated groups, respectively. Results are means ± s.e.m. ** *p* < 0.01, *** *p* < 0.01 compared to Cnr1^−/−^ mice (**a**,**b**); * *p* < 0.05, ** *p* < 0.01 compared to vehicle (**c**,**d**).

**Figure 2 molecules-26-05089-f002:**
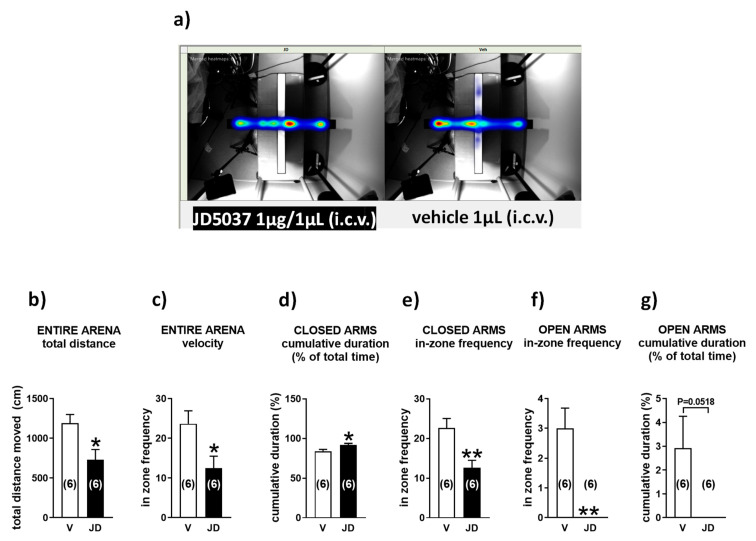
Intracerebroventricular microinfusion of JD5037 increases anxiety-related behavior in the elevated plus maze test. Heat maps (**a**) and summary of mouse activities in individual compartments (**b**–**g**) of the elevated plus maze. Animals received i.c.v. infusion of JD5037 (1 µg; JD) or its vehicle (3% DMSO, 8% Tween 80, 30% PEG-400, 59% saline; V). They were tested in the elevated plus maze 1 h later. The computerized EthoVision video tracking system was used for data collection and analysis. Bars are mean ± s.e.m. * *p* < 0.05, ** *p* < 0.01, compared with vehicle.

**Figure 3 molecules-26-05089-f003:**
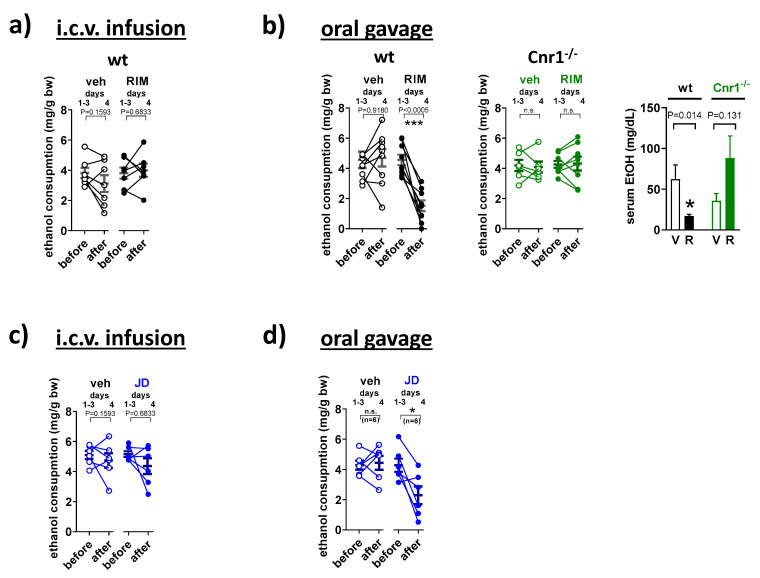
Relative involvement of central vs. peripheral CB1R blockade in the inhibition of voluntary ethanol intake by CB1 antagonists. Mice had access to 20% ethanol for 4 h each day. On day 4, one hour before the dark period, mice were infused intracerebroventricularly (i.c.v.) with (**a**) rimonabant (2 µg, RIM, R), (**b**) JD5037 (1 μg, JD) or their solvents (veh, V) and drinking session was repeated one more time. Another cohort of wild-type mice (wt) and/or their CB1 receptor-deficient (Cnr^−/−^) counterparts received (**c**) rimonabant (10 mg/kg), (**d**) JD5037 (3 mg/kg), or vehicle by oral gavage. Drinking behavior in individual animals is expressed as points before (average of days 1–3) and after treatment (day 4). The corresponding serum ethanol values are shown as mean ± s.e.m. * *p* < 0.05; *** *p* < 0.001, compared with before treatment (Student’s *t*-test for paired samples), ^#^ *p* < 0.05 compared with the vehicle (Student’s *t*-test for unpaired samples), n.s. not significant.

**Figure 4 molecules-26-05089-f004:**
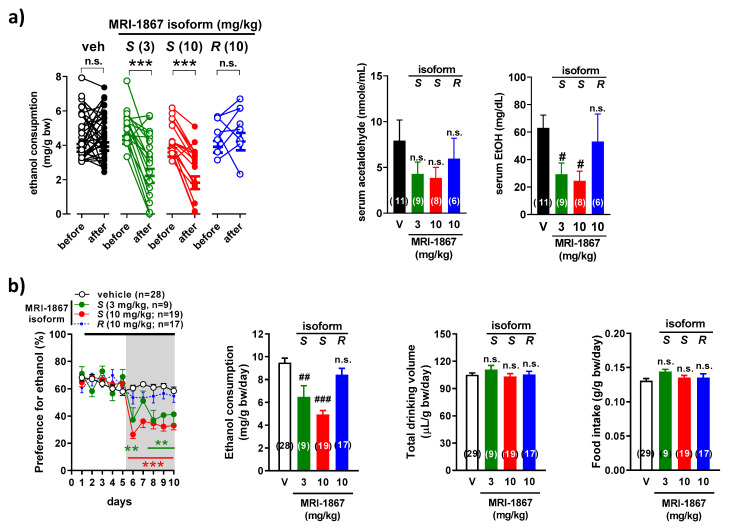
MRI-1867 decreases alcohol consumption after oral administration in two drinking models. (**a**) C57BL/6J mice had access to 20% ethanol for 4 h daily. On day 4, one hour before the dark period, mice received *S*-1867, (3, 10 mg/kg; S), *R*-MRI-1867 (10 mg/kg; R), or vehicle (V) by oral gavage and drinking session was repeated. Serum level of acetaldehyde and alcohol from blood obtained at the end of the drinking session. (**b**) Mice had free access to a 15% ethanol solution and water, using a two-bottle free-choice paradigm. From days 6 to 10, mice received daily *S*-MRI-1867 (3, 10 mg/kg; S), *R*-MRI-1867 (10 mg/kg; R), or vehicle (V) by oral gavage. Drinking behavior in individual animals (**a**) is expressed as points before (average of days 1–3) and after treatment (day 4). Other points and bars (**a**,**b**) are mean ± s.e.m. of daily to 5-day drinking behavior, respectively. ** *p* < 0.01, *** *p* < 0.001, compared with before treatment (Student’s *t*-test for paired samples) (**a**) or with vehicle (two-way ANOVA followed by Tukey’s multiple comparisons test) (**b**); ^#^ *p* < 0.05, ^##^ *p* < 0.01, ^###^ *p* < 0.001 compared to vehicle (one-way ANOVA followed by Dunnett’s post hoc test), n.s. not significant.

**Figure 5 molecules-26-05089-f005:**
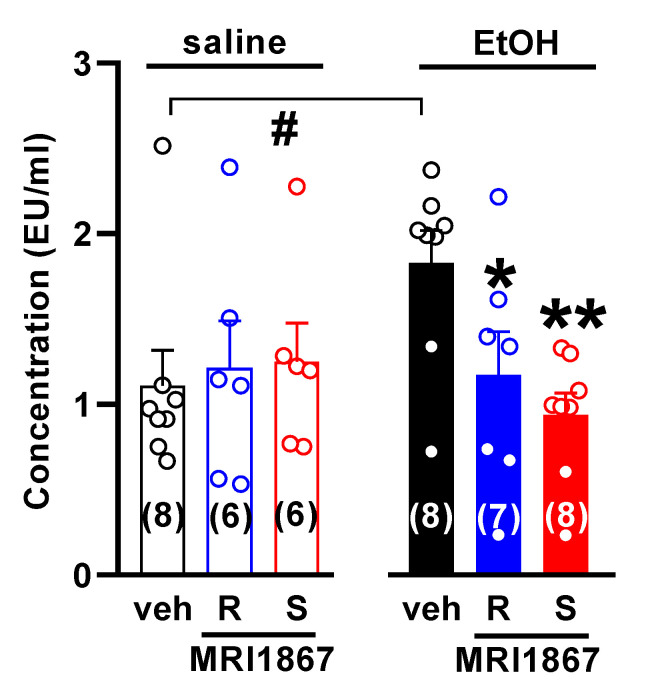
The iNOS inhibitor *R*-MRI-1867 decreases serum endotoxin level in acutely alcohol-intoxicated mice. Mice received *S*-MRI-1867 10 mg/kg (S) or *R*-MRI-1867 10 mg/kg (R) or vehicle (veh) by oral gavage (time 0), followed by intragastric administration of 6 g/kg ethanol (30% *w*/*v*; EtOH) or saline at 30 min. Endotoxin was measured in the serum obtained from the portal vein 1 h after the acute alcohol challenge. Results are expressed as mean ± s.e.m. * *p* < 0.05, ** *p* < 0.01, compared with vehicle in alcohol-treated group (one-way ANOVA followed by Dunnett’s post hoc test), ^#^ *p* < 0.05, compared with saline-treated mice.

**Table 1 molecules-26-05089-t001:** Concentration of *S*-MRI-1867 in different segments of the gastrointestinal tract.

Tissue	Concentration (µmol/g Wet Tissue Weight)
stomach	30.57 ± 8.05
duodenum	36.88 ± 11.63
jejunum	44.14 ± 9.18
ileum	52.78 ± 11.50
colon	17.62 ± 5.66

## Data Availability

Not applicable.

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
