# Peer review of "Effects of a Peripherally Restricted Hybrid Inhibitor of CB1 Receptors and iNOS on Alcohol Drinking Behavior and Alcohol-Induced Endotoxemia"

_molecules, 2021, doi:10.3390/molecules26165089_

Round 1

Reviewer 1 Report

In this manuscript, the authors have shown that the voluntary alcohol intake was inhibited upon systemic administration of CB1R antagonists, including the peripherally restricted hybrid CB1R antagonist/iNOS inhibitor S-MRI-1867. They also investigated that both the R- and S-MRI-1867 effectively inhibited alcohol-induced increase in blood endotoxin. Overall, the findings of the paper are interesting and of high importance in context of peripheral CB1R to alcohol consumption behavior. The inhibition effect of CB1R antagonists to central vs peripheral CB1R was well shown in the mice model with applicable dataset and appropriate experiments. I suggest that the manuscript should be accepted with some minor modification/correction.

There are a few concerns about the manuscript.

  1. Please indicate both the full and abbreviation name of “c.v” (line 88), “i.p.” (line 102) and “BAC” (line 152) where they first appeared in the main text.
  2. Use uniform font size for all the figures, including the x- and y-axis labels, text, and symbols. Especially, some labels in figure 3 were too small to be distinguished. While the labels in figure 5 were too large.
  3. Line 113: “Mice routinely let go of the bar within 1 second”. But one of the times was 1.6 ± 0.9 s.
  4. Line 116: “Another 30 min passed before the hypothermic (a) and cataleptic (b) responses were measured.” This description was not clear. Because Figure 1a included both hypothermic and cataleptic, but they were not labeled in figure 1b.
  5. Line 119: “*P<0.05; **P<0.01, ***P<0.01 compared to vehicle or Cnr1-/- mice; n=5.” In this sentence, please give the correct representation for “***”.
  6. Figure 2: Please label each graph consistently with other figures (Figure 2a, 2b, 2c, etc). And describe them properly in the main text. Avoid use “upper panels” or “bottom panels” (line 127, 137 and 140).
  7. Line 130-133: “Figure 2. This is a figure ……” The sentences need to be revised or removed.
  8. Figure 3d: as indicated in figure 3b, it is necessary to perform the control experiments with Cnr-/- mice treated with JD5037.
  9. Figure 4: The statistical significance was not explicitly marked on some of the column figures, serum acetaldehyde change (figure 4a), etc. Even if the results are not significant, it should be marked as “ns” to avoid confusion.
  10. Figure 5: Bracket the n values.

Author Response

Response to Reviewer 1 Comments

Dear Reviewer,

Thank you for taking the time to evaluate our work. We appreciate your thoughtful comments that helped improve the quality of our work. We did our best address your remarks in the manuscript text. Our changes have been marked up with the “Track Changes” in the revised manuscript version and were detailed below in red, following every comment of the Reviewer.

  1. Please indicate both the full and abbreviation name of “v” (line 88), “i.p.” (line 102) and “BAC” (line 152) where they first appeared in the main text.

Response 1: Full names and abbreviations were provided in the manuscript text according to the Reviewer’s instructions.

  1. Use uniform font size for all the figures, including the x- and y-axis labels, text, and symbols. Especially, some labels in figure 3 were too small to be distinguished. While the labels in figure 5 were too large.

Response 2: All figures, including Fig 3 and Fig 5 were re-formatted and simplified to make them more transparent, legible, and consistent. Changes were also reflected in legends to figures.

  1. Line 113: “Mice routinely let go of the bar within 1 second”. But one of the times was 1.6 ± 0.9 s.

Response 3: The indicated value “1.6 ± 0.9 s” was an unfortunate addition not associated with the study. It was, therefore, removed from the revised text. Furthermore, to avoid unspecific information, legend was modified in a following way: ‘Mice routinely held the bar for 0.3±0.3s and 0.3±0.3s in wt and ko groups, respectively’. We apologize for the confusion.

  1. Line 116: “Another 30 min passed before the hypothermic (a) and cataleptic (b) responses were measured.” This description was not clear. Because Figure 1a included both hypothermic and cataleptic, but they were not labeled in figure 1b.

Response 4: Figure 1 was re-labeled as four (4) separate graphs (a, b, c, d) to better reflect different experimental conditions. Legend was corrected accordingly.

  1. Line 119: “*P<0.05; **P<0.01, ***P<0.01 compared to vehicle or Cnr1-/- mice; n=5.” In this sentence, please give the correct representation for “***”.

Response 5: The representation for “ *** ” was provided in the manuscript. Additionally, as the labeling of graphs changed (a, b, c, d instead of a and b), so did the of description of statistical analysis. Thus, panels (a-b) and (c-d) were described separately in the legend text to Figure 1.

  1. Figure 2: Please label each graph consistently with other figures (Figure 2a, 2b, 2c, etc). And describe them properly in the main text. Avoid use “upper panels” or “bottom panels” (line 127, 137 and 140).

Response 6: Figure 2 graphs were labeled according to the Reviewer’s suggestions. The appropriate changes were also made in the legend lines (127-131), Material and Methods (lines 357-32) and in the main text (lines 136-151). Please note that the sequence of panels has also changed to better reflect the text flow.

  1. Line 130-133: “Figure 2. This is a figure ……” The sentences need to be revised or removed.

Response 7: The line was removed from the manuscript text. It was a leftover of the of the journal’s template on which the manuscript was built. We apologize for causing the confusion.

  1. Figure 3d: as indicated in figure 3b, it is necessary to perform the control experiments with Cnr-/-mice treated with JD5037.

Response 8: We could not agree more with the Reviewer about the need of additional experiments. Unfortunately, we are unable to address this point as we do not have enough mutant mice available for the study right away. We are, so to speak, in the mercy of our “CB1 heterozygous breeders”, which do not always comply with our needs. It may take a few months before generating enough KO and WT mice for this experiment. This far exceeds the 5-day timeframe we were given to address the points.

We kindly ask the Reviewer to consider an alternative solution, which is the addition of the following statement/explanation to our manuscript text: “Our earlier observation indicates that this effect of JD5037 was CB1R-depended and did not occur in CB1R-deficient mice [33].”  

We think the additional experiments with JD5037 would mostly replicate our finding from Cell Metab, 2019. 29(6): p. 1320-1333 e8 (Godlewski et al., ref #33 in the manuscript), which showed that CB1 deficient mice were insensitive to oral administration of JD5037 after oral administration in the two-bottle-choice test.

  1. Figure 4: The statistical significance was not explicitly marked on some of the column figures, serum acetaldehyde change (figure 4a), etc. Even if the results are not significant, it should be marked as “ns” to avoid confusion.

Response 9: Figures and legends were updated to show more explicitly the statistical information requested by the Reviewer.

  1. Figure 5: Bracket the n values.

Response 10: The ’n’ values were bracketed as indicated by the Reviewer.

The manuscript also underwent an extensive English revision as part of the professional service offered by native English-speaking personnel of the NIH Library. The following edits were implemented:

  • Abstract, line 21: the expression ‘among others’ replaced with ‘among other things’
  • Abstract, line 23: Quotes added to the following expressions: “two-bottle” as well as a “drinking in the dark”
  • Abstract, line 25: the abbreviation ‘i.c.v.’ was replaced with a full word ‘intracerebroventricular’
  • Introduction, line 53: the article ‘the’ was removed making the sentence read as follows: ‘once in circulation’ (instead of ‘once in the circulation…’)
  • Results, line 285: a comma was added making the sentence read as follows: ‘Hermansky-Pudlak syndrome, pulmonary fibrosis…..’.
  • Materials and Methods, lines 359-361: the sentence was modified as follows: ‘ The position of bottles was changed every day and alcohol and water bottles were replaced every 4 days ‘.
  • Abbreviations for intracerebroventricular injections were made consistent in the manuscript text e.g., ‘i.c.v.’ instead of icv (see lines 314 and 335).

I hope these improvements are satisfactory and will meet your approval criteria.

Kind regards,

Grzegorz Godlewski

Reviewer 2 Report

This manuscript aims to evaluate the role of peripheral CB1 receptors on alcohol intake in mice. Experiments using a non-brain penetrant CB1 receptor antagonist inhibited alcohol intake and a peripherally restricted hybrid CB1R antagonist/iNOS inhibitor reduced alcohol consumption as well, supporting the predominant role of peripheral CB1 receptors in the regulation of alcohol consumption. This manuscript gives strong evidence of the role of peripheral CB1 receptors in alcohol consumption providing support for future therapeutic use of the simultaneous inhibition of CB1 receptors and iNOS.

Minor comments

  1. Delete “Figure 2. This is a figure. Schemes follow another format. If there are multiple panels, they should be listed as: (a) Description of what is contained in the first panel; (b) Description of what is contained in the second panel. Figures should be placed in the main text near to the first time they are cited. A caption on a single line should be centered.”, in lines 130-133.
  2. In line 273 the word “that” is written twice.
  3. Review the sentence in lines 317-318 “The infusion rate and volume were controlled the syringe pump”
  4. The sentence in lines 359-360 should be written in past tense.

Author Response

Response to Reviewer 2 Comments

Dear Reviewer,

Thank you for taking the time to evaluate our work. We appreciate your thoughtful comments that helped improve the quality of our work. We did our best address your remarks in the manuscript text. Our changes have been marked up with the “Track Changes” in the revised manuscript version and were detailed below in red, following every comment of the Reviewer.

  1. Delete “Figure 2. This is a figure. Schemes follow another format. If there are multiple panels, they should be listed as: (a) Description of what is contained in the first panel; (b) Description of what is contained in the second panel. Figures should be placed in the main text near to the first time they are cited. A caption on a single line should be centered.”, in lines 130-133.

Response 1: The line was removed from the manuscript text. It was a leftover of the of the journal’s template on which the manuscript was built. We apologize for causing the confusion.

  1. In line 273 the word “that” is written twice.

Response 2: The repetition has been deleted.

  1. Review the sentence in lines 317-318 “The infusion rate and volume were controlled the syringe pump”

Response 3: The incomplete sentence has been corrected as follows: “The infusion rate and volume were controlled through the use of the syringe pump (Harvard Apparatus PHD 22/2000, USA).”

  1. The sentence in lines 359-360 should be written in past tense.

Response 4: The sentence has been corrected and it reads as follows: “The position of bottles was changed every day and alcohol and water bottles were replaced every 4 days.”

 The manuscript also underwent an extensive English revision as part of the professional service offered by native English-speaking personnel of the NIH Library. The following edits were implemented:

  • Abstract, line 21: the expression ‘among others’ replaced with ‘among other things’
  • Abstract, line 23: Quotes added to the following expressions: “two-bottle” as well as a “drinking in the dark”
  • Abstract, line 25: the abbreviation ‘i.c.v.’ was replaced with a full word ‘intracerebroventricular’
  • Introduction, line 53: the article ‘the’ was removed making the sentence read as follows: ‘once in circulation’ (instead of ‘once in the circulation…’)
  • Results, line 285: a comma was added making the sentence read as follows: ‘Hermansky-Pudlak syndrome, pulmonary fibrosis…..’.
  • Materials and Methods, lines 359-361: the sentence was modified as follows: ‘ The position of bottles was changed every day and alcohol and water bottles were replaced every 4 days ‘.
  • Abbreviations for intracerebroventricular injections were made consistent in the manuscript text e.g., ‘i.c.v.’ instead of icv (see lines 314 and 335).

I hope these improvements are satisfactory and will meet your approval criteria.

Kind regards,

Grzegorz Godlewski

Reviewer 3 Report

The manuscript by Luis Santos-Molina et. al. reports the involvement of peripheral cannabinoid receptor 1 (CB1R) in the control of alcohol drinking behavior in mice and the potential use of a hybrid molecule that acts as both an inverse agonist of peripheral CB1R and an inhibitor of inducible nitric oxide synthase to combat alcohol drinking behavior and alcohol-induced endotoxemia.

Minor points

  1. Please add the Animal Experimentation permit number.
  2. What was the reason for using 15% ethanol in the 2-bottle choice test and 20% ethanol in the Drinking in the Dark test?
  3. The legend to Figure 2; lines 130-133. Please delete »Figure 2. This is a figure. Schemes follow an-130 other format. If there are multiple panels, they should be listed as: (a) Description of what is contained in the first panel; 131 (b) Description of what is contained in the second panel. Figures should be placed in the main text near to the first time 132 they are cited. A caption on a single line should be centered«.
  4. The use of abbreviations should be consistent throughout the manuscript, e.g. icv or i.c.v.
  5. Lines 455-456; Values are presented as mean ± s.e.m., with the number of replicates and the level of significance reported in figures, figure legends and supplementary tables. Please delete »and supplementary tables« as the manuscript does not contain supplementary data.
  6. Concentration of S-MRI-1867 in different segments of the gastrointestinal tract are reported in µM (Table 1). In the Material and Methods sections (lines 436-437), the authors state that the values are expressed as nmol/mL in wet tissue weight.

Author Response

Response to Reviewer 3 Comments

Dear Reviewer,

Thank you for taking the time to evaluate our work. We appreciate your thoughtful comments that helped improve the quality of our work. We did our best address your remarks in the manuscript text. Our changes have been marked up with the “Track Changes” in the revised manuscript version and were detailed below in red, following every comment of the Reviewer.

  1. Please add the Animal Experimentation permit number

Response 1: The permit number has been added in line 307 and the sentence reads as follows: “All animal procedures were approved by the Institutional Animal Care and Use Committee of NIAAA, NIH (Animal Experimentation permit number LPS-GK-1), and the experiments were carried out in accordance with its guidelines.”

  1. What was the reason for using 15% ethanol in the 2-bottle choice test and 20% ethanol in the Drinking in the Dark test?

Response 2: In the current study, we used two models that were adequately described in our earlier paper (Godlewski et al., Cell Metab, 2019. 29: 1320-1333; referred to here as ref #33). These models differ with respect to whether the animal had a choice between alcohol and water or was exposed to only one bottle at a time. In the drinking in the dark paradigm, the animal is exposed only a single (alcohol) bottle for a short time (4 hours at night). This short-term drinking is meant to reproduce a human behavior (binge drinking). It is reflected by high blood alcohol concentration. In contrast, when animal has a choice (e.g., the two-bottle parading) it will prefer 12-15% alcohol concentration over water. Higher concentrations of alcohol in the bottle (e.g., 20%) would rather make the animal approach water bottle more often. It would be difficult to measure animal’s preference when alcohol concentration exceeds 15%. We added the following statements to help understand the two different paradigms by Readers:

For drinking in the dark paradigm (lines 157-158): “Mice exposed to 20% alcohol for a short period at night tend to drink to inebriation, reflected by high blood levels of ethanol [33]”.

For the two bottle choice test (lines 186-188) sentence was modified as follows: “Consistent with our earlier study [33], male C57Bl6/J mice with continuous access to water or and 15% ethanol solution displayed high preference for alcohol (64.2 ± 1.2 %)…..”.

  1. The legend to Figure 2; lines 130-133. Please delete »Figure 2. This is a figure. Schemes follow an-130 other format. If there are multiple panels, they should be listed as: (a) Description of what is contained in the first panel; 131 (b) Description of what is contained in the second panel. Figures should be placed in the main text near to the first time 132 they are cited. A caption on a single line should be centered«.

Response 3: The line was removed from the manuscript text. It was a leftover of the of the journal’s template on which the manuscript was built. We apologize for causing the confusion.

  1. The use of abbreviations should be consistent throughout the manuscript, e.g. icv or i.c.v.

Response 4: As indicated by the Reviewer, abbreviations were corrected for consistency e.g., ‘i.c.v.’ instead of icv (see lines 314 and 335).

  1. Lines 455-456; Values are presented as mean ± s.e.m., with the number of replicates and the level of significance reported in figures, figure legends and supplementary tables. Please delete »and supplementary tables« as the manuscript does not contain supplementary data.

Response 5: The sentence was corrected and reads as follows: “Values are presented as mean ± s.e.m., with the number of replicates and the level of significance reported in figures and figure legends”.

  1. Concentration of S-MRI-1867 in different segments of the gastrointestinal tract are reported in µM (Table 1). In the Material and Methods sections (lines 436-437), the authors state that the values are expressed as nmol/mL in wet tissue weight.

Response 6: The units in which we presented tissue levels of MRI-1867 was rather unfortunate and confusing, and we apologize for making it look like the drug concentration in the solution. The correct format should be in µmol/g wet tissue weight. It did not affect the calculated values, only the units in which they were expressed. Units were corrected throughout the text e.g., Discussion (line 214), description of Table 1 (line 217), Material and Methods (lines 455-456).

I hope these improvements are satisfactory and will meet your approval criteria.

Kind regards,

Grzegorz Godlewski
